# Multi Locus View: an extensible web-based tool for the analysis of genomic data.

Martin J. Sergeant [1], Jim R. Hughes [1,2], Lance Hentges [1], Gerton Lunter[1,3], Damien J. Downes [2] & Stephen Taylor [1✉]

Tracking and understanding data quality, analysis and reproducibility are critical concerns in the biological sciences. This is especially true in genomics where next generation sequencing (NGS) based technologies such as ChIP-seq, RNA-seq and ATAC-seq are generating a flood of genome-scale data. However, such data are usually processed with automated tools and pipelines, generating tabular outputs and static visualisations. Interpretation is normally made at a high level without the ability to visualise the underlying data in detail. Conventional genome browsers are limited to browsing single locations and do not allow for interactions with the dataset as a whole. Multi Locus View (MLV), a web-based tool, has been developed to allow users to fluidly interact with genomics datasets at multiple scales. The user is able to browse the raw data, cluster, and combine the data with other analysis and annotate the data. User datasets can then be shared with other users or made public for quick assessment from the academic community. MLV is publically available at https://mlv.molbiol.ox.ac.uk.

[1] MRC WIMM Centre for Computational Biology, MRC Weatherall Institute of Molecular Medicine, University of Oxford, Oxford, UK. [2] MRC Molecular Haematology Unit, MRC Weatherall Institute of Molecular Medicine, University of Oxford, Oxford, UK. [3] University Medical Centre Groningen, Department of Epidemiology, University of Groningen, Groningen, The Netherlands. ✉email: stephen.taylor@imm.ox.ac.uk

Next generation sequencing (NGS) technologies such as ChIP-seq, RNA-seq and ATAC-seq generate vast amounts of data, which, once mapped, is analysed with programs such as MACS[1] and DESeq2[2] to extract biologically meaningful signals. The final output of these pipelines is usually a list of genomic regions filtered by an enrichment level or fold change and a statistical threshold, such as *p*-value, *q*-value or FDR. Selecting thresholds in absence of the ability to effectively see their effect on the final dataset can lead to the loss of biologically meaningful signal or the inclusion of noise and common bioinformatic mapping artefacts depending on the stringency used.

Understanding the effectiveness of the parameters used for a given dataset, data type or analytical tool is extremely challenging and effective quality control of such outputs may require the user to manually go through tens of thousands of regions to validate that the chosen thresholds, which is extremely time consuming in traditional genome browsers. Importantly, with the advent and high impact of machine learning in the genomics field, there is a critical need for a platform to generate curated high quality training sets of genomic regions which match specific criteria. Although many excellent genome browsers exist for looking at genomic locations, such as the UCSC genome browser[3], the WashU Epigenome Browser[4], IGV[5] and HiGlass[6], these are designed for sequential visualisation of specific individual loci of interest, rather than looking at an experiment as a whole.

Multi Locus View (MLV) allows rapid filtering of hundreds of thousands of locations based on their metadata, combined with the genome views of regions of interest. MLV additionally allows for interaction with the complete dataset via the use of a highly customisable range of interactive charts fully linked up with the embedded genome browser. Going beyond data interaction MLV also provides the ability to run commonly required procedures, such as intersection between genomic annotations, but also advanced analyses, such as dimensionality reduction. This provides a powerful and easy to use way to discover new insights and quality control large 'omics data sets. We demonstrate the power of MLV by showing examples of false positive inclusion with ENCODE datasets and characterisation of enhancers and promoters in a large published dataset[7]. Importantly, MLV only requires BED and bigWig tracks as an input, which unlike BAM files, are lightweight but extremely flexible and information rich summaries of the data that allow for extremely fast and fluid interactions with complete datasets.

## Results

**Identification of ChIP-seq false positives with MLV**. ChIP-seq experiments are performed to identify sites of chromatin modification or protein binding, visualised as peaks. For genome wide analysis, peaks are identified bioinformatically, most commonly with MACS2, though newer methods use digital signal processing[8] and Machine Learning[9]. Often these peak callers use arbitrary statistical thresholds which can lead to inclusion of false positives, affecting downstream analysis. To demonstrate the ability of MLV to filter true- and false-positive peaks, we looked at data from H3K27ac ChIP-seq (a marker of active transcription) in the human prostate cancer cell line 22Rv1 (ENCSR391NPE). We uploaded 46,030 MACS2 peak calls and associated bigWig files to MLV (see Supplementary Methods 1). Sorting by -log$_{10}$ q-value showed that 12,000 of the identified peaks (26%), those with a value less than 10, were little more than background noise (Fig. 1). This demonstrates the ability of MLV to visually interrogate peak calling results and complement statistical analysis by determining an appropriate q or p value threshold for filtering high quality peak calls. This allows for easy and intuitive

segregation of the basic analysis into strong or weak peaks, producing stringent or more generous annotation, quickly and on the fly.

**Functional annotation of regulatory elements**. The genome contains three main types of regulatory elements (promoters, enhancers and boundaries) whose position can be detected in open chromatin assays such as DNase-seq[10] and ATAC-seq[10,11]. Open chromatin is common to all these classes and so cannot determine a specific element's identity alone. However epigenetic marks (H3K4me1, H3K4me3, H3K27ac) and transcription factor binding (CTCF) have been used to annotate elements, such as likely promoters and enhancers using the ratio of H3K4me1 to H3K4me3[7]. To demonstrate how MLV can be used to fluidly explore and classify all the open chromatin elements in a given cell-type, we used chromatin marks to cluster and annotate erythroid open chromatin peaks identified from ATAC-seq based on their relative enrichment of ChIP-seq signals (see Supplementary Methods 2 and Fig. 2). Filtering for peaks with a high H3K4me1 to H3Kme3 ratio (Fig. 2c(iii)), characteristic of enhancers, identifies a distinct cluster with few peaks overlapping TSSs, low levels of CTCF and a range of H3K27ac - expected traits of enhancer regulatory elements (Fig. 2b(ii), (iv), (v)). Conversely, putative promoter peaks with a low H3K4me1 to H3K4me3 ratio showed a high proportion of TSS overlap (61%), and higher levels of both H3K27ac and CTCF, again confirming the expected traits of promoters (Fig. 2c). The tagging functionality could then be used to append classes to each class of open chromatin region. Using MLV we were able to quickly and efficiently categorise, and annotate peaks, whilst filtering out actual peaks from background noise. These peaks can then be exported for use in downstream statistical analysis such as motif discovery or nearest gene analysis.

**Analysis of cohesin/CTCF interactions**. The 3D structure of the genome is thought to be mediated via the cohesin complex and CTCF, which bring distal regions of DNA together via a process of loop extrusion[12]. In a recent paper[13], the authors mutated CTCF such that it abolished interaction with SCC1, a member of the cohesin complex, in the human HAP1 cell line. They then carried out ChIP-seq of CTCF and SCC1 in both the wild type and the mutant cells. This revealed that in the main CTCF binding to DNA was unaffected in the mutant, whereas cohesin (SCC1) was reduced, especially at locations where CTCF was also bound, hence supporting the hypothesis that CTCF stabilizes cohesin on chromatin. To explore these datasets dynamically, the peak locations and bigWig tracks from the ChIP-Seq experiments were loaded into MLV along with histone mark data from Hap1 cells (see Supplementary Methods 3 and Fig. 3).

A Histogram of CTCF peak height fold change (Fig. 3a(ii)) generally supported the paper's observation that the mutation did not reduce CTCF chromatin binding. Indeed, the shape of the histogram indicated an average log2 fold change value of around 0.5, suggesting the mutation may cause a modest increase in CTCF binding. The cohesin (SCC1) histogram (Fig. 3a(ii)) showed a skew to the left indicating reduced binding in the mutant and this region contained mainly locations with high CTCF binding. Again, this was in accordance with the paper's findings that in the CTCF mutant, cohesin binding was generally reduced, especially at locations where CTCF was also bound. Further exploration of data was carried out by selecting those locations with 'high' CTCF (see Supplementary Methods 3) but where SCC1 binding in the mutant did not decrease (Fig. 3b(ii)) These regions showed a bi-modal distribution in CTCF change (Fig. 3b(i)). The smaller peak mainly contained black listed

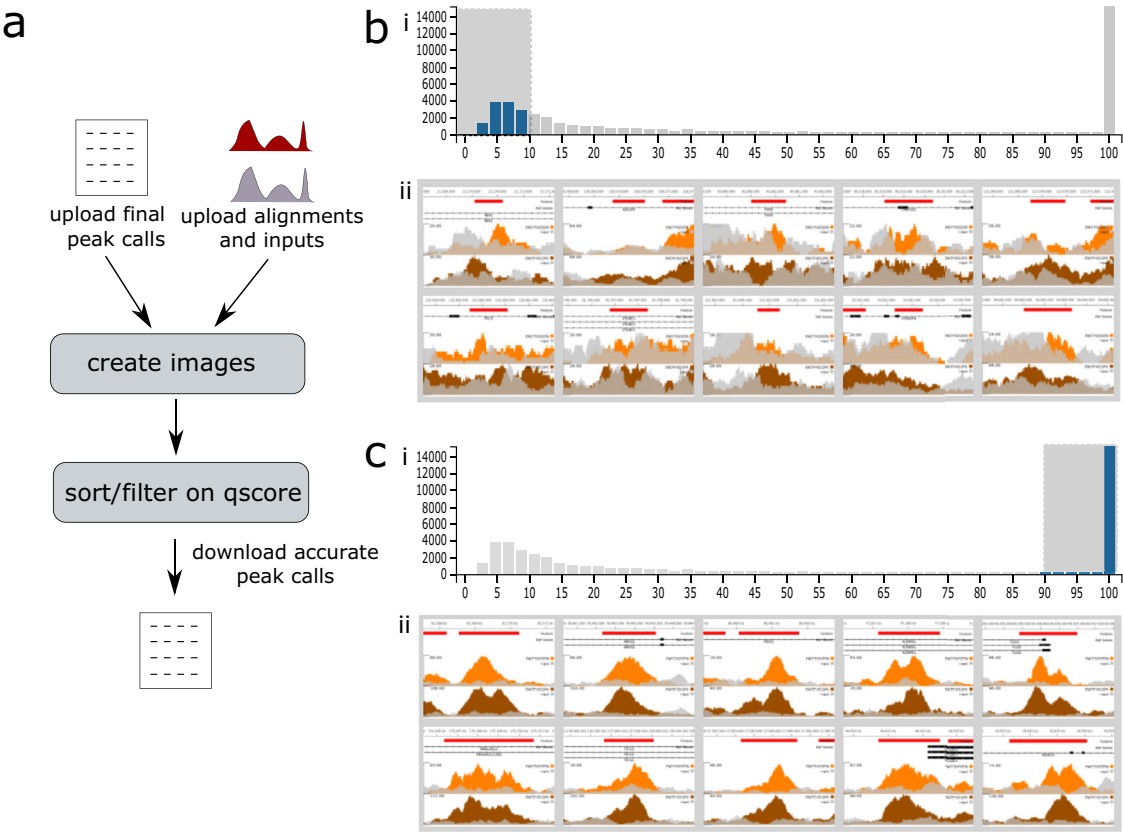

**Fig. 1 Summary of peaks in encode project ENCSR391NPE.** https://mlv.molbiol.ox.ac.uk/projects/multi_locus_view/1434 **a** Workflow (see supplementary methods 1 for full details). **b** Locations with $-\log_{10} q$ values less than 10 have been selected using the histogram (**i**), clearly showing regions (**ii**) which are misidentified as peaks. **c** Locations with high $-\log_{10} q$ values selected (**i**), showing genuine peaks (**ii**). The two alignment tracks ENCFF025ZEN and ENCFF421QFK are orange and brown respectively and the corresponding input tracks, ENCFF483ELD and ENCFF769UET are grey.

regions and indeed images from these locations showed these areas with the abnormal peak structure associated with the bioinformatic artefacts found in these regions (Fig. 3c(i)). However, the larger histogram peak represented regions containing what looked like genuine peak calls (Fig. 3c(ii)). Moreover, the SCC1 peaks in these regions exhibited a different pattern, being broader and flatter than the majority of peaks at other locations (Fig. 3c(iii)) and, in many cases, the corresponding CTCF peak was at the edge of the cohesin peak. These peaks also appeared to be enriched for promoters due to their association with TSSs and regions that have greater levels of H3K4em3 (Supplementary Fig. 2). Such regions could be marked in MLV and exported into RStudio or a Jupyter notebook to check their statistical significance. The above analysis shows the ease with which MLV can be used to explore important published data, to both confirm the basic findings and to add extra insights. as was exhibited by the discovery of a putative class of CTCF binding elements associated with promoters.

## Discussion

The massive expansion in NGS data generation and the increasing complexity of datasets and data types makes it difficult to interpret and validate the results without referring back to the underlying data. To tackle this challenge requires better ways of analysing and humanly interacting with such large multidimensional datasets. Importantly, such methods should have very low barriers to use, not requiring specialised computational skills and so allowing for their general use in the biological community. Similarly, they need to have quick, fluid, and above all intuitive interfaces to allow researchers to concentrate on

asking the pertinent biological questions rather than on the computational tasks required to ask them.

MLV is able to complement existing statistical packages in the following ways. Firstly, results containing p values from programs such as MACS2, can be visually examined to see if they are biologically meaningful in the context of the experiment. This may help with selecting an appropriate cut off. Secondly, if certain patterns become evident whilst visualising the data, regions can be appropriately annotated and the data exported to packages in R and Python to ascertain whether they are statistically significant.

MLV provides a more holistic way of interacting with complex NGS data sets. By combining the use of common lightweight data formats (e.g. BED, bigWig and tab delimited text) with a fully featured JavaScript frontend with powerful server-side Python flask frameworks and PostgreSQL relational databases, MLV provides a powerful and agile web-based interface to complex datasets. The inbuilt and dynamically linked dimensionality reduction functionality, table, graph and image based interfaces within MLV allows a user to simultaneously analyse the dataset as a whole and also quickly drill down to subsets with specific characteristics or behaviours. By allowing for the clustering and subsequent fine grained inspection of multiple genome regions of similar characteristics in a single view, MLV affords a powerful way to look for trends in results data traditionally represented in tables and static figures. Also with MLVs dynamic filtering, graphing and data brushing, it is possible to visually inspect and understand the effects of parameter selection. Importantly, to aid transparency, the data, analysis and visualisations can be shared via online URL to provide a powerful supplement to any

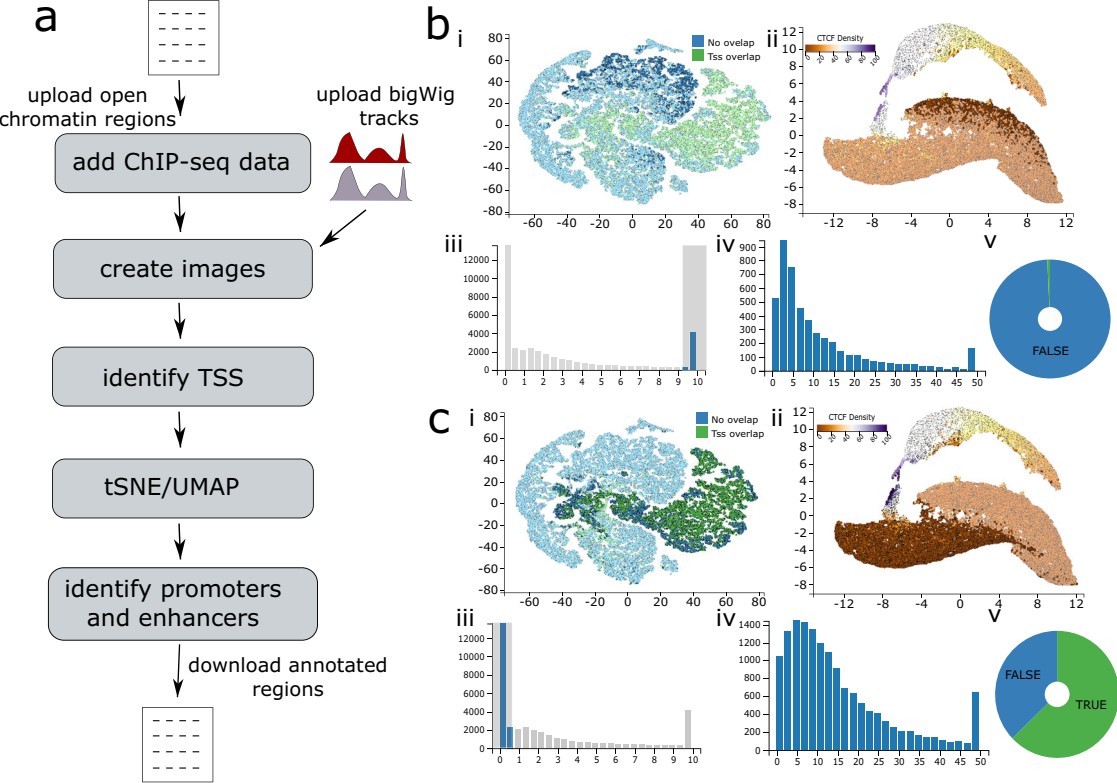

**Fig. 2 Functional annotation of regulatory elements.** https://mlv.molbiol.ox.ac.uk/projects/multi_locus_view/1590. **a** Workflow (see supplementary methods 2 for full details) (**b** and **c**) show the clustering of putative enhancers and promoters respectively, based on enrichment for the two chromatin marks and CTCF binding. (**i**) t-SNE targeting the density of all peaks, with green points showing regions which overlap with TSSs. The plot shows the large enrichment for overlapping annotated TSSs with the H3K4me3 enriched cluster. (**ii**) UMAP targeting the density of all peaks. The plot is coloured by CTCF peak density using a PuOr11 colour scale, from brown (most dense) to purple (least dense), identifies clusters of strongly and weakly bound elements. (**iii**) Histogram of the H3K4Me1/H3K4m3 ratio of open chromatin sites allows for the interactive selection of differentially enriched elements. (**iv**) Histogram of H3K27ac enrichment allows for the interactive selection of elements most enriched for this active chromatin mark. (**v**) Dynamically linked pie charts show the enrichment of a given annotation (e.g., TSS overlap, green segment) for the selected or filtered objects.

publication, which gives the reader or manuscript reviewer immediate access to the datasets and analysis that underpins the findings of the work. Such interfaces will be critical to enable greater transparency and reproducibility in research. Furthermore, we have shown that data files or analyses released with published datasets can be rapidly incorporated into MLV to allow for the re-exploration of existing data and analyses to validate the conclusions of the manuscript, to discover new trends in the data or to ask new biological questions with the inclusion of further datasets or annotations.

Finally, the intrinsic ability to cluster data, or to input clustered data, combined with the fine grained visualisation and ability to tag collections of data points means MLV can quickly generate extremely large high-quality training datasets for machine learning approaches. The generation of such validated training sets is extremely laborious on the operator and so represent the biggest bottleneck to the wide-scale implementation of these powerful methods in genomics research.

## Methods

MLV takes as input, a tsv or csv file, where the first three columns specify the genomic location and an unlimited number of additional columns containing metadata for that location. Examples include a simple BED file, the output of MACS2[1] or an Excel file that has been saved in csv or tsv format. The data can then be combined with annotations (e.g. transcription start sites [TSS]) and sequencing data (e.g. bigWig files). Dimension reduction (UMAP, tSNE) can also be carried out to identify clusters. In addition dynamic graphs, genomic tracks and images can be added to further aid visualisation/analysis. After sorting and filtering,

locations can be annotated (tagged) and then exported and links generated for sharing, such as with reviewers or with a publication. Figure 4 shows a general summary of MLV and the functions available to the user

**Visualisation.** The main view consists of three panels, a spreadsheet-like table housing the genomic locations and all the metadata, a genome browser and a panel showing dynamic graphs/charts. All three panels are linked - data can then be filtered by selecting regions/sections on a graph or using the table, which instantly updates the other graphs, table and browser.

The spreadsheet contains the genomic locations and any associated metadata. In addition, if images have been generated, these can be displayed either as thumbnails in the spreadsheet or as rows in their own table. This allows instant visualisation of common elements in filtered data sets. When displayed on their own, images are sorted in the same order as the table rows and can have their border coloured according to any field in the data, further aiding the elucidation of patterns. Clicking on a row/image will display the genomic location in the browser and highlight its position on any scatter graphs present. Extra columns can also be generated by applying simple arithmetic to existing columns, For example, calculating the ratio between the signals in two bigWig tracks will produce a column that can be sorted by relative enrichment or depletion between the two datasets. Columns can also be deleted if the data is no longer required.

The internal browser is a lightweight JavaScript component based on igv.js (https://github.com/igvteam/igv.js). Initially the internal browser displays a gene track (if a genome was specified) and a track showing the uploaded genomic features. To aid visualisation, these features can be coloured by any of the metadata fields, positioned along the y-axis proportional to a numeric field and labelled with another field. Only those features that are in the current filter are displayed and clicking on a feature will highlight the relevant image/row in the table and highlight the appropriate point on any scatter plots present. Tracks of common formats. BAM, bigWig, bigBed, tabix indexed bed.gz files etc. and UCSC browser sessions can be added to the browser. In addition, many of the analysis steps will also add tracks. Thumbnails for every location (displayed in the table panel) can be generated based upon the current browser tracks and settings.

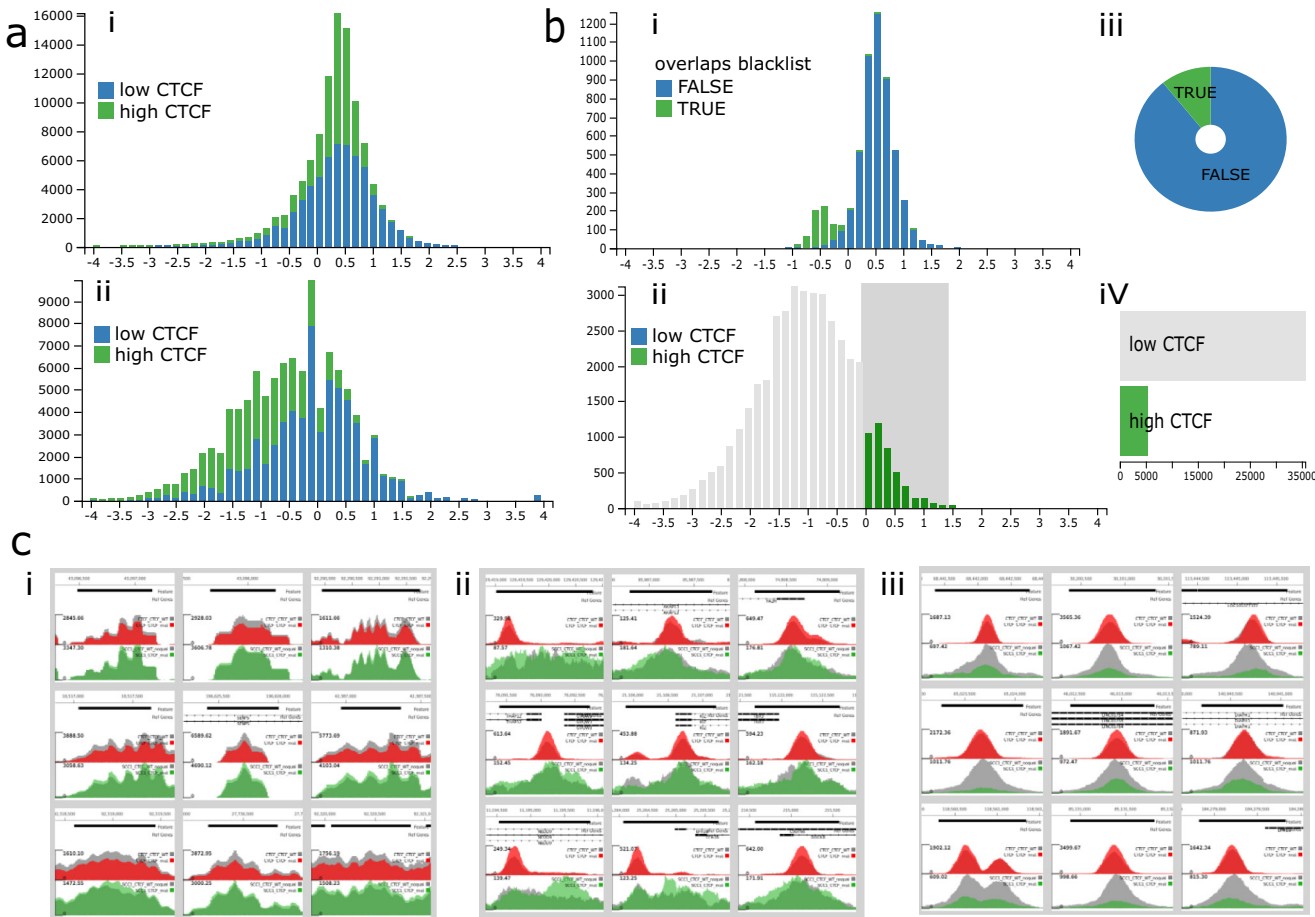

**Fig. 3 Analysis of SCC1 and CTCF ChIP-seq data.** https://mlv.molbiol.ox.ac.uk/projects/multi_locus_view/2057 Data taken from from the The Structural Basis for Cohesin-CTCF-anchored Loops[11]. **a** Histograms of CTCF (**i**) and cohesin (**ii**) log2 fold changes. The bars are coloured by tags (green - high CTCF, blue - low CTCF). **b** Filtering of strong CTCF binding sites with no decrease in SCC1 binding in the CTCF mutant. (**i**) Histogram showing the bimodal distribution of the CTCF log2 fold changes between the mutant and the WT in the selected regions. The green bars show black listed regions. (**ii**) Histogram of cohesin log2 fold change between the mutant and WT, the grey box shows the regions (those with a log2 fold change greater than 0) that were selected. (**iii**) Pie chart showing the proportion of black listed regions (green) in the selected regions. (**iv**) Row chart showing tags, which was used to select regions with strong CTCF binding (high CTCF). **c** Representative samples of genome browser images. The upper track shows CTCF ChIP-seq peaks with the grey track being WT and the red track, the CTCF mutant. The lower track shows SCC1 (cohesin) data with the WT grey and the CTCF mutant and green (**i**) the small left-hand peak in **b**(**i**) consisting of black listed regions, (**ii**) regions with strong CTCF binding, but no reduction of cohesin binding in the CTCF mutant (the large right-hand peak in **b**(**ii**)), (**iii**) typical peaks for the majority of regions with strong CTCF binding.

Graphs including scatter plots, histograms, box plots and pie charts can be added to the view, showing any of the metadata fields that are appropriate to the graph. All graphs respond to filtering, and can also be used to intuitively filter the data by selecting appropriate regions. For example, selecting regions near TSS sites on a histogram will also update a histogram displaying H3K4me3 ChIP-seq peak data, showing an overall increase in height. Another example would be selecting regions on a UMAP/t-SNE generated scatter plot, which updates graphs displaying fields that were used to generate the clustering, therefore indicating which fields influence different clusters. As well as users being able to add graphs and adjust their settings, appropriate ones are automatically generated in many of the analysis methods.

Although the application contains a browser which will display each selected view, it is often more informative to visualise many locations at once, in order to see if there is commonality in a filtered data set. To this end images based on the browser view can be generated for every location. Once all images are created, they can be viewed as thumbnails as part of a table row or in their own table.

**Analysis methods**. A number of methods can be employed to help interpret the data. Most methods will add fields (columns) to the table, graphs and tracks to the browser aiding the interpretation of the data. If a genome was specified, then the nearest Transcription Start Site (TSS) based on the RefSeq annotation will be calculated using bedtools[14]. The RefSeq id and common name of the gene is also given and there is an option to include molecular function Gene Ontology (GO) annotations[15]. These annotations were simplified by first obtaining all GO terms for a RefSeq gene by using gene2go and gene2refseq files from NCBI. Next, using

the go-basic.obo file from http://geneontology.org/, multiple terms for each gene were further expanded by traversing up the hierarchy and adding terms at each level. Then at each hierarchical level, terms were collapsed by only keeping the most frequent, resulting in a much simplified scheme, where each RefSeq gene had a single term at each hierarchical level. Users can choose to include up to five levels of GO annotations in the data returned from a TSS search.

In order to fully interpret some data sets, it is usually useful to combine the existing data with other datasets. Hence MLV enables the user to intersect locations with a list of genomic regions or the locations in other projects. This feature uses bedtools[14] behind the scenes and also allows information contained in the intersecting data to be added to the data set, easily allowing other experiments to be incorporated into the project.

BigWig files from experiments such as ChIP-Seq can be specified and MLV will calculate the area, max height of the track's signal for each genomic location. This data can then be used to plot various graphs or used to generate UMAP/t-SNE scatter plots (see below).

In order to group the genomic locations it may be useful to cluster them based on specified fields. For example, clustering based on ChIP-Seq peaks for histone marks may give an indication of promoters/enhancers. MLV enables this, by using the dimension reduction algorithms, UMAP[16] and t-SNE[17] implemented with scikit-learn[18]. Any number of numeric fields (columns) can be used as input and these are reduced to a specified number (default 2) of output dimensions. Although the initial graphs show the first two dimensions for each algorithm, the user can produce different graphs by mixing and matching dimensions from different algorithms.

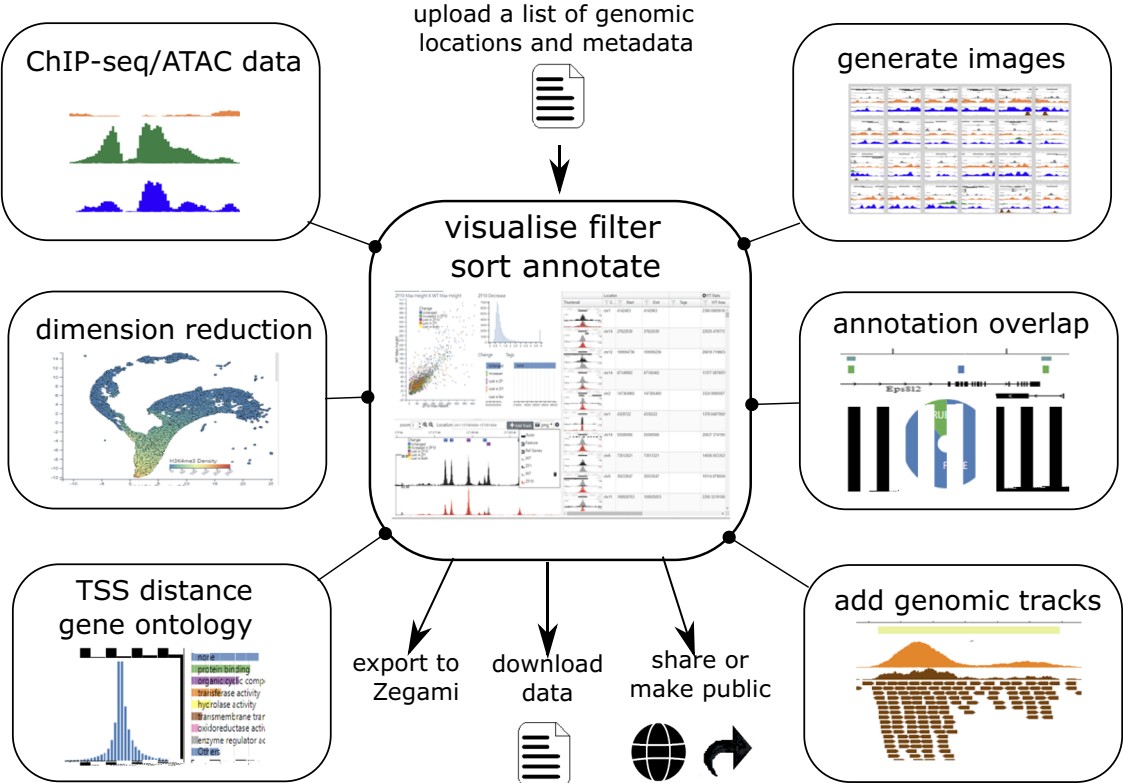

**Fig. 4 Summary of MLV Workflow.** Initially a BED or BED-like file of genome locations is uploaded, which can be visualised via a spreadsheet, browser and various interactive graphs. A number of analyses can also be carried out on the uploaded locations in order to further explore the data. These include calculating distances from genomic objects such as TSSs, overlap with genomic features, calculating signal data at each location from bigWig e.g. ChIP or ATAC-seq and dimensionality reduction values, to produce interactive t-SNE[17] or UMAP[16] representations of the data. To aid interpretation, genome tracks of other datasets and dynamic graphs can be added and browser images for each location can be generated, which can be viewed in a table format. The analysis can be downloaded as csv/tab delimited text or shared/made public via a URL. A collection can also be created in the analysis software Zegami for further interrogation of the data.

**Output options**. In order to communicate your findings with others, annotation of each location is possible. Tasks such as naming clusters, marking outliers/ anomalies etc. can be quickly achieved by assigning tags to filtered sets. In addition, individual images/table rows or ranges can be tagged. The data can be downloaded as a tsv or csv file and current settings (graphs and browser layout) can be saved and the view shared with other users, either with or without edit permissions or the project can be made shared via URL such that even non users will be able to view it. The whole data or filtered subsets can be cloned to produce new data sets. Moreover, if images have been generated it can be exported to the visual data exploration software Zegami (https://zegami.com/) for further analysis.

**Implementation and extensibility**. The backend of MLV is implemented using the python framework flask (http://flask.pocoo.org/) and the relational database PostgreSQL (https://www.postgresql.org/). It is composed of two main building blocks: projects (analysis types) and jobs (pipelines), see supplementary Figure 1. It was designed to be modular, with each independent module specifying the analysis types (projects) and jobs (pipelines) required. In addition to MLV, two other modules have been developed, LanceOtron (https://lanceotron.molbiol.ox.ac.uk/) that calls peaks using machine learning and CaptureSee (http://capturesee.molbiol. ox.ac.uk/[19]) for looking at highly multiplexed Capture C data[20]. The front end is written in JavaScript and is built upon two stand-alone components, MLVPanel (https://github.com/Hughes-Genome-Group/MLVPanel) and CIView (https:// github.com/Hughes-Genome-Group/CIView). MLVPanel is a lightweight, extensible genome browser, based on igv.js (https://github.com/igvteam/igv.js), but with the emphasis on displaying multiple genomic locations simultaneously. All tracks are displayed compactly on the same canvas and many panels can be displayed on the same web page, for example as thumbnails in a table. It is highly extensible, making it simple for developers to create their own custom tracks to suit the needs of a project and a node.js version allows images (png, svg or pdf) to be created programmatically. CIView enables users to intuitively look at multivariate data, visualising the effect that each parameter has on the dataset as a whole. It is based upon dc charts (https://dc-js.github.io/dc.js/), which in turn uses d3 (https://d3js. org/) and crossfilter (https://square.github.io/crossfilter/). However, to address the problem of the slow rendering of large graphs in SVG which can only cope with a few thousand points, the default scatter plots have been replaced with those using WebGL technology and thus is able to cope with hundreds of thousands of data points. It also allows users to dynamically create and manipulate graphs in order to tailor the display to their dataset. Graphs can either be linked to a spreadsheet (https://github.com/mleibman/SlickGrid), giving the ability to edit the data or a table displaying images, enabling instant visual feedback at each filtering step.

**Reporting summary**. Further information on research design is available in the Nature Research Reporting Summary linked to this article.

## Data availability
All data in the examples is available for download from MLV, with the relevant links in the figure legends.

## Code availability
The code is freely available on GitHub https://github.com/Hughes-Genome-Group/mlv under the GNU General public license.

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

## Acknowledgements

This work was carried out as part of the WIGWAM Consortium (Wellcome Investigation of Genome Wide Association Mechanisms) funded by a Wellcome Trust Strategic Award (106130/Z/14/Z), Medical Research Council (MRC) Core Funding (MC_UU_00016/14) and with support from the National Institutes of Health (USA) grant number R24DK106766 and from the Alan Turing Institute grant number AfSMBPP1\100037.

## Author contributions

J.R.H. and S.T. conceived the initial idea. M.J.S. wrote the code. J.R.H., S.T. and M.J.S. wrote the manuscript. L.H., D.J.D., J.R.H., G.L. and S.T. tested the software and suggested new features.

## Competing interests

The authors declare the following competing interests: S.T. is founder and CSO of Zegami, J.H. is founder of Nucleome Therapeutics and G.L. is co-founder of Genomics plc. The remaining authors declare no competing interests.
