## [Peer Review File · Communications Biology]

Reviewers' Comments:

Reviewer #1:

None

Reviewer #2:

Remarks to the Author:

Seargeant et al. present a webserver-based service facilitating visualization and summarization of genomic datasets. Supported are standard input file formats (bigwig, bed, csv) for data generated with NGS techniques such as ChIP-seq, RNA-seq and ATAC-seq. The application allows summarization of arbitrary numeric as well as categorical columns in the input files, and visualization in forms of colour-coded histograms, barplots or pie charts. A filtered version of the input data can be downloaded. An interesting feature is simultaneous display of genomic tracks for multiple genomic regions from the input file ("multilocus view").

The multilocus view is an innovative idea that has the potential to facilitate quality control of genomic datasets, replacing the otherwise tedious viewing of single regions in a standard genomic browser. The summary plots for columns of the input file may be particularly useful to non-commandline users and provide a nice enhancement of functionality. The application examples illustrated in the paper represent standard modern-day approaches to genomic data analysis. The server instance at <https://mlv.molbiol.ox.ac.uk> is user-friendly and responsive. The graphical interface is quite carefully designed, the interactive features are working well and can be used intuitively. Two example sessions are available for viewing.

I have the following remarks to the authors:

I have tested the webserver with 18.04.1-Ubuntu and Firefox 80.0.1 (64-bit) .

1. I wasn't able to find a detailed documentation specifying requirements on the input files. Importantly, it wasn't clear that the server only accepts ucsc-style chromosome names and will fail on ensembl-style (integer) chromosome names. Concretely, uploading a bed file with ensembl chromosome names (standard in my institution) failed. Also uploading a bigwig file produced with deepTools failed complaining about rows containing unequal numbers of columns.

I'd suggest to the authors to either provide a visible link to detailed documentation on input file requirements on the server landing page, or to relax these requirements such that files of different provenance can be uploaded.

2. Related to point 1, clicking on the "Help" button forwarded me to an empty page with an unformatted header.

I'd suggest to the authors to fix this issue, or to point the user to a FAQ/Troubleshooting section.

As the upload of local files failed for me, I have further tested the server with the example sessions available on the landing page.

3. As the authors point out in the manuscript, data analysis reproducibility and transparency is an important aspect of publication. For this matter, a processed dataset is only interpretable together with detailed metadata describing parameter values and operations applied to it.

In that sense, I wasn't able to find this information in either of the example sessions on the server. Information on the underlying organism, genome version, and annotation version was not available from within the session. A history of operations applied to the dataset was also missing e.g. to allow for reproducing a particular plot.

I'd suggest to the authors to track important metadata as well as user history, and make it available for download to the user, or for sharing in another form. Similarly, as for example the Galaxy platform does it.

4. I wasn't able to upload an additional bigwig file to one of the example sessions, nor to save the session as my own project.

I'd suggest to the authors to add this functionality, that extends sharing of static sessions to building upon other users' sessions, again, similar to what the Galaxy platform offers.

5. Using the "forward" and "backward" browser buttons resulted in resetting of the example session to the initial state. I wasn't able to find a way to bookmark the latest session state.

I'd suggest to the authors to add this functionality, if otherwise not available in the user-defined datasets.

6. Currently, the server is only available as a web service at one location.

Do the authors have plans on making it portable such that users could run their own local instances ?

Reviewer #3:

Remarks to the Author:

The MLV tool is a well-built browser that has the potential to provide a large impact in the computational biology community. MLV achieves numerous milestones in the area of genomics visualization that are unprecedented. The first milestone is flexibility without programming experience: users can upload and manipulate data in MLV without requiring any programming experience. The second milestone is visualization of multiple genomic loci in an intuitive manner that does not overwhelm the user. This is achieved by the ability to summarize data with various charts, and the generation of multiple genome browser thumbnails that can be easily traversed. The third milestone is responsiveness: many existing genome browsers are slow and unable to handle data that is provided. MLV seamlessly handles large tabular data without, to my knowledge, stalling or crashing. The attached review contains minor comments that address concerns of overall tool support, documentation, and how certain points in the paper are addressed.

I hope you find these comments useful.

Best,

Alyssa Morrow

Rebuttal Letter

Reviewer #2

(1) I wasn't able to find a detailed documentation specifying requirements on the input files.

There is now an information box on the upload page explaining what type of file is required and the information is also in the documentation (Input File Required)

Importantly, it wasn't clear that the server only accepts ucsc-style chromosome names and will fail on ensembl-style (integer) chromosome names. Concretely, uploading a bed file with ensembl chromosome names (standard in my institution) failed.

The server will now accept UCSC and Ensembl style chromosome names (explained on the upload page and in the documentation).

Also uploading a bigwig file produced with deepTools failed complaining about rows containing unequal numbers of columns.

As MLV is genome region based Bigwig files may not be uploaded as the initial track (see Input File Required) but may be visualised within the MLV browser (see Adding Tracks).

BigWig files can be added later, in order to calculate peak area/height at each location (see Calculate Peak Stats).

I'd suggest to the authors to either provide a visible link to detailed documentation on input file requirements on the server landing page, or to relax these requirements such that files of different provenance can be uploaded.

Information about which files can be uploaded is on the landing page and the requirements for UCSC style chromosome names have been relaxed(see Input File Required).

(2) Related to point 1, clicking on the "Help" button forwarded me to an empty page with an unformatted header. I'd suggest to the authors to fix this issue, or to point the user to a FAQ/Troubleshooting section.

The help button in the navigation bar now allows you to email a question or go to the main documentation page.

(3) As the authors point out in the manuscript, data analysis reproducibility and transparency is an important aspect of publication. For this matter, a processed dataset is only interpretable together with detailed metadata describing parameter values and operations applied to it.

In that sense, I wasn't able to find this information in either of the example sessions on the server. Information on the underlying organism, genome version, and annotation version was not available from within the session.

This information can now be obtained by clicking the information icon in the toolbar.

A history of operations applied to the dataset was also missing e.g. to allow for reproducing a particular plot. I'd suggest to the authors to track important metadata as well as user history, and make it available for download to the user, or for sharing in another form. Similarly, as for example the Galaxy platform does it

Similar to Galaxy, MLV now records users actions which can be viewed in the history dialog. This shows what the user did, e.g. added graphs/tracks, ran an analysis job etc. Also, similar to Galaxy, actions can be deleted, which deletes the consequences of the action i.e. the charts, tracks, data columns that were added by the action.

(4) I wasn't able to upload an additional bigwig file to one of the example sessions, nor to save the session as my own project. I'd suggest to the authors to add this functionality, that extends sharing of static sessions to building upon other users' sessions, again, similar to what the Galaxy platform offers.

Projects made public, such as the example one, are static and cannot be changed (to avoid unspecified users altering your figures). You can however, add graphs/tracks as well as alter existing ones in order to better explore the data. You can also clone the project (using "Save As") or create a subset from it (see Permissions), which allows you to build upon the session. In addition, you can share projects with other people and give them editing rights.

(5) Using the "forward" and "backward" browser buttons resulted in resetting of the example session to the initial state. I wasn't able to find a way to bookmark the latest session state.

I'd suggest to the authors to add this functionality, if otherwise not available in the user-defined datasets.

Session state is automatically saved when adding charts/tracks/columns and running analysis jobs. This can now be reversed in the history dialog (see above in point 3). In addition, the overall layout e.g. graph/tracks size and colors, table column order/width etc. can be saved at any time (see Saving a Project).

(6) Currently, the server is only available as a web service at one location.

The code is freely available on github (<https://github.com/Hughes-Genome-Group/mlv>) along with documentation about how to install it:-

https://mlv.readthedocs.io/en/latest/mlv_developer/developer.html

Reviewer #2

Results: Identification of ChIP-seq false positives with MLV

(1) The authors use this analysis to convey how MLV can be used to choose parameters for data processing by filtering ChIP-seq peaks by a cutoff. However, this analysis does not convey why MLV is the best way to choose cutoffs, opposed to filtering data which a more traditional methods, such as in R or python. How do the visualizations and functionality in MLV provide particular benefit for making decisions regarding data cutoffs?

The strength of MLV is that filters can be applied quickly by a user friendly GUI and visualized interactively. By selecting an area on a graph the effects on other parameters can instantly be visualized in the other graphs and charts, as well as seeing the underlying biological signals in all the filtered genomic locations. For example, it is possible to see the shape of all the peaks in a ChIP-Seq experiment interactively which allows the biologist to choose a sensible cut off rather than a default. Often such cutoffs or thresholds are applied

without actually looking at the underlying data which leads to poor quality peak calls being accepted for further analysis or publication.

Furthermore, constructing easy to use and powerful user interfaces is time consuming and MLV is very useful for R and Python developers who do not have expertise or the time to construct these for communicating results. This allows them to focus on generating results tables and loading them into MLV. These results may then be shared with biologists who may help spot problems with the data which increases collaboration and explainability of the results.

(2) In Figures 2B and 2C, it is not clear if the authors produce side-by-side browser subfigures in MLV or if these subfigures were combined outside of MLV. It is possible these figures were generated using thumbnail visualizations in MLV, but this feature should be highlighted in this figure so users know how these subfigures were generated.

Figures 2B and 2C were combined outside of MLV. Browser and chart images can be separately downloaded from MLV. In our experience, users usually want the individual images/figures and arrange them in their own way outside of the application.

Functional annotation of regulatory elements.

(3) This analysis claims that tSNE was run in MLV. However, I could not find how to run tSNE. The methods section should explicitly state how views for all analyses in this manuscript were produced.

The supplemental methods now explicitly explains step by step how the views (ENCODE project ENCSR391NPE, Functional annotation of regulatory elements and Analysis of cohesin/CTCF interactions) were created with links to the relevant parts of the documentation. In addition, projects now have a history, where each step can be viewed to show the user how the view was created.

(4) In Figures 3Bi, and 3Ci, it is not clear which variables are targeted in dimensionality reduction. These variables should be stated in the caption.

The figure legend and supplemental methods (Functional annotation of regulatory elements) now show that the peak area/max height at each location were used in the dimension reduction.

Analysis of cohesin/CTCF interactions

(5) Figures 4A and 4B, should state what the fold change is comparing and what the positive and negative values represent.

The figure caption now states the fold changes are between mutant and WT and it is the log2 transformation hence the negative values.

(6). One issue with using MLV instead of statistical packages is that you are relying on the untrained eye to make conclusions. For example, the paper states that Figure 4Aii shows that the cohesin fold change is shifted to the left. However, it could be a possibility that given p-values, these results are not significant. Is it possible to incorporate statistical significance into these plots?

With MLV, you are looking for trends based on the visual analysis that could potentially be missed with automated statistical pipelines. These can then be followed up if significance is sought. However, statistical tests within MLV is an excellent idea and as more people use the software we will get a better feedback of what type of tests are most useful and the best way to present the results.

Technical issues Upon using MLV, I came across some minor technical issues that should be addressed.

(7). When trying to register for MLV, it failed with the error “Email is already in use” but then sent me an email to create an account.

This has not been reported by other users, but of course if I get any complaints I will look into this..

(8). My browser (Chrome, MACOSX) does not allow me to horizontally scroll in the data table. Therefore, I was unable to visualize or access a majority of the data. This is a high severity technical problem.

We have Mac users and this problem has not been reported. I was unable to replicate this problem when testing on a Mac but would be happy to follow up if more details could be given.

9. MLV is very sensitive to screen size, and some figures were cut off or pushed out of screen on a smaller screen. There should be recommendations of which screen devices MLV works best on, or it should use adaptive screen size techniques.

MLV works best with larger screens, which allows users to observe how linked tracks, graphs, and images change when values are filtered.. The size of each section can be altered with the slider, individual graphs can be moved and resized, and tracks can be reordered and their height adjusted in order to get the exact view required. Adaptive screen size techniques would just show individual panels/graphs on small screen sizes thus greatly limiting the functionality of the application. You can change the size of each panel using the dividers. In the documentation I have explained this and suggested using the largest screen size as possible

10. When zooming out of the tSNE plot in

https://mlv.molbiol.ox.ac.uk/projects/multi_locus_view/1590, the data disappeared and the “Reset all” button did not bring the data back. I was able to go to “Settings, Centre Plot” to re-center the data but this was hard to find.

This was bug and thanks for reporting. Zooming on the scatter plot has been made less sensitive and the ‘Reset All’ will now also center any scatter plots..

11. I was unable to find documentation for MLV. The MLV interface has an overwhelming amount of options. For users to understand the tool, documentation should address the following points:

- a) How a user can set up, save, and share sessions
- b) Which file formats can be visualized in MLV
- c) Instructions on where users can submit issues or questions
- d) Tutorials that walk through how the analyses in the results section were set up

Documentation is located here

https://mlv.readthedocs.io/en/latest/multi_locus_view/multi_locus_view.html

(a) set up, save, and share sessions, Creating a Project, Sharing a Project

(b) Which file formats can be visualized in MLV Input File Required

(c) Submitting issues and questions Submitting an Issue

(d) There are three tutorials which can be accessed on the home page which show how the example projects were created:

<https://youtu.be/rfa2os4237Q>, <https://youtu.be/7jNnIPHR6pE>, <https://youtu.be/55Vx-JDPchw>

REVIEWERS' COMMENTS:

Reviewer #2 (Remarks to the Author):

The authors have now addressed all the revision comments and implemented fixes and extensions in their software, as well as provided extensive documentation. Testing of the MLV web server was now successful.

MLV server is a promising tool with the potential to facilitate generation and visualisation of summary statistics for multiple genomic loci in parallel, using NGS-standard input files. It may facilitate NGS data reanalysis and promote reproducibility.

Reviewer #3 (Remarks to the Author):

I thank the authors for their careful responses to my initial review. Most of my concerns and comments have now been clarified. One of my main concerns in the initial submission were regarding documentation. However, the authors have now provided detailed documentation and videos describing how to use MLV. I believe that these resources are sufficient.

My main remaining concern refers to comment (6)* in the initial review. This comment mentions that MLV relies on the untrained eye to make conclusions when, in some instances, a statistical analysis would be a better way to make conclusions. This was pointed out in Figure 4, where detecting changes in distributions visually lacks statistical rigor for such analyses. I think that MLV is a very useful tool for use cases such as those described in Figures 3. However, the authors should be careful not to suggest that visual analysis is a comparable substitution for statistic tests. In this light, it would be good to mention how MLV can complement or enhance analyses that can currently be run in existing exploratory environments that support statistical analysis, such as Rstudio or a python notebook. Users of MLV will most likely use a combination of MLV and existing tools to analyze a given dataset. For this reason, highlighting interplay between existing tools is important to help the user understand how MLV can fit in to their current pipelines.

*Comment (6)

(6) One issue with using MLV instead of statistical packages is that you are relying on the untrained eye to make conclusions. For example, the paper states that Figure 4Aii shows that the cohesin fold change is shifted to the left. However, it could be a possibility that given p-values, these results are not significant. Is it possible to incorporate statistical significance into these plots?

Response To Reviewers Comments

Reviewer #1

Major Comments

I have tried revisiting MLV and trying it out using my own data but unfortunately I haven't been able to (not sure if this is just me because clearly others have managed to make it work). I attached the screenshot of my issue.

The problem was impossibly large numbers in the chromosome coordinates of the uploaded file, which was not detected by any of the initial parsing. During processing of the file it caused an error and led to a cryptic error message for the user. We have since replaced this with a more sensible message. We encourage the reviewer to get in contact with us directly if he/she wishes further help with their dataset.

Minor Comments

1) Results, first paragraph: "This demonstrates the ability of MLV to visually interrogate peak calling results and determine an appropriate rather than an arbitrary threshold for filtering high quality peak calls."

Do they authors claim that visually inspecting peaks to determine parameter selection is more effective, or a better approach to statistical methods? Visual inspection does not indicate statistical significance.

The visual inspection of peaks is often used to verify data sets, to train models and test data for comparing peak calling algorithms (Rye et al. 2017, Hocking et al. 2011) though it is not often feasible to do so. However, we are not suggesting visually inspecting peaks as an alternative to statistical methods but complementing it. It provides a sanity check on the results and may help in selecting an appropriate p/q value cutoff e.g 0.05 or 0.01 with which to filter the peaks in order to get more meaningful results in downstream analysis. The above sentence has been changed to:-

'This demonstrates the ability of MLV to visually interrogate peak calling results and complement statistical analysis by determining an appropriate q or p value threshold for filtering high quality peak calls'

2) Page 4: "Using MLV we were able to quickly and efficiently categorize, and annotate peaks, whilst rapidly inspecting specific and random peaks to quality check the annotations - the entire analysis taking less than an hour."

What does "rapidly inspecting" signify. I presume a trained user could "rapidly" inspect peaks and perform "quality checking of the annotations", and the whole thing taking less than an hour, but this measure of time and efficiency is highly subjective and shouldn't be portrayed in such a way as to lead users into thinking "this is how long it should take". I also do not understand the difference between "specific and random peaks".

We have removed the reference to time and replaces 'specific and random peaks' to 'actual peaks and background noise' :-

Using MLV we were able to quickly and efficiently categorize, and annotate peaks, whilst filtering out actual peaks from background noise. These peaks can then be exported for use in downstream statistical analysis such as motif discovery or nearest gene analysis.

3) Page 6: *"although incomplete normalization of the BigWig tracks may also account for this." Why didn't the authors download generate normalized BigWigs themselves to eliminate this possibility?*

We did download and generate the bigwig files using deeptools and per kilobase per million reads normalization (information in supplementary method 3). Therefore, we have removed the phrase *'incomplete normalization of the BigWig tracks may account for this'*

4) Page 6: *"This shows the ease and facility with which MLV can be used to explore important published data, to both confirm the basic findings and to add extra insights." I was not entirely clear on which part of the analysis added extra insights.*

An example of this is when we analyse the novel class of CTCF binding elements associated with promoters, where loss / interaction between CTCF and SCC1(cohesin) does not prevent cohesin binding. The sentence has been changed to emphasise this *'and to add extra insights as was shown by the discovery of a putative novel class of CTCF binding elements associated with promoters.'*

5) Discussion: *"The massive expansion in NGS data generation and the increasing complexity of datasets and data types has led to the current crisis in both the transparency of interpretation and reproducibility of data in the biomedical sciences." Not sure that I would call it a crisis, I would like to think that with workflow management systems, code repositories, research reproducibility and transparency has improved. This is a subjective statement unless supported by some other published work. Please revise this statement.*

The sentence has been changed to *'The massive expansion in NGS data generation and the increasing complexity of datasets and data types makes it difficult to interpret and validate the results without referring back to the underlying data'*

Reviewer #2

Remarks to the Author:

The authors have now addressed all the revision comments and implemented fixes and extensions in their software, as well as provided extensive documentation. Testing of the MLV web server was now successful.

MLV server is a promising tool with the potential to facilitate generation and visualisation of summary statistics for multiple genomic loci in parallel, using NGS-standard input files. It may facilitate NGS data reanalysis and promote reproducibility.

No actions required

Reviewer #3:

Remarks to the Author:

I thank the authors for their careful responses to my initial review. Most of my concerns and comments have now been clarified. One of my main concerns in the initial submission were regarding documentation. However, the authors have now provided detailed documentation and videos describing how to use MLV. I believe that these resources are sufficient.

My main remaining concern refers to comment (6) in the initial review. This comment mentions that MLV relies on the untrained eye to make conclusions when, in some instances, a statistical analysis would be a better way to make conclusions. This was pointed out in Figure 4, where detecting changes in distributions visually lacks statistical rigor for such analyses. I think that MLV is a very useful tool for use cases such as those described in Figures 3. However, the authors should be careful not to suggest that visual analysis is a comparable substitution for statistical tests. In this light, it would be good to mention how MLV can complement or enhance analyses that can currently be run in existing exploratory environments that support statistical analysis, such as Rstudio or a python notebook. Users of MLV will most likely use a combination of MLV and existing tools to analyze a given dataset. For this reason, highlighting interplay between existing tools is important to help the user understand how MLV can fit into their current pipelines.*

We agree, MLV was never meant to replace statistical tests, but rather complement them.

We see its use as providing a sanity check on the output of automated pipelines and may be helpful in selecting appropriate p/q value cutoffs, which can alter depending on the nature of the underlying data. This is explained in the first example:-

'This demonstrates the ability of MLV to visually interrogate peak calling results and complement statistical analysis by determining an appropriate α or p value threshold for filtering high quality peak calls'

Also trends or patterns that are discovered by visually inspecting the data, which may have been missed by automated pipelines, should be annotated and exported to statistical packages to check their validity. This is explained in the second example:-

'Using MLV we were able to quickly and efficiently categorize, and annotate peaks, whilst filtering out actual peaks from background noise. These peaks can then be exported for use in downstream statistical analysis such as motif discovery or nearest gene analysis.'

And the third example:-

Such regions could be marked in MLV and exported to Rstudio or a python notebook to check their statistical significance.

In addition we have added a paragraph to the discussion:-

MLV is able to complement existing statistical packages in the following ways. Firstly, results containing p values, from programs such as MACS2, can be visually examined to see if they are biologically meaningful in the context of the experiment. This may help with selecting an appropriate cut off. Secondly, if certain patterns become evident whilst visualising the data, regions can be appropriately annotated and the data exported to packages in R and Python to ascertain whether they are statistically significant.

References

Rye, M. B., Sætrom, P. & Drabløs, F. A manually curated ChIP-seq benchmark

demonstrates room for improvement in current peak-finder programs. *Nucleic Acids Research* vol. 39 e25–e25 (2011).

Hocking, T. D. *et al.* Optimizing ChIP-seq peak detectors using visual labels and supervised machine learning. *Bioinformatics* **33**, 491–499 (2017).